# The Application of Infrared Thermography in the Assessment of BEMER Physical Vascular Therapy on Body Surface Temperature in Racing Thoroughbreds: A Preliminary Study

**DOI:** 10.3390/ani14111538

**Published:** 2024-05-23

**Authors:** Karolina Nawrot, Maria Soroko-Dubrovina, Paulina Zielińska, Krzysztof Dudek, Kevin Howell

**Affiliations:** 1Institute of Animal Breeding, Wroclaw University of Environmental and Life Sciences, Chelmonskiego 38C, 51-160 Wroclaw, Poland; 112604@student.upwr.edu.pl; 2Department of Surgery, Wroclaw University of Environmental and Life Sciences, pl. Grunwaldzki 51, 50-366 Wroclaw, Poland; paulina.zielinska@upwr.edu.pl; 3Faculty of Mechanical Engineering, Wroclaw University of Science and Technology, 50-370 Wroclaw, Poland; krzysztof.dudek@pwr.edu.pl; 4Microvascular Diagnostics, Royal Free Hospital, Pond Street, London NW3 2QG, UK; kevin_howell@btinternet.com

**Keywords:** physical vascular therapy (BEMER), infrared thermography, thoroughbreds, body surface temperature, vascularity

## Abstract

**Simple Summary:**

Infrared thermography (IRT) is a fast and non-invasive physiological diagnostic method that allows highly accurate monitoring of body surface temperature. Detected temperatures are variable and change according to blood flow regulation at the skin surface. Many physical therapies, including the pulsed electromagnetic field system BEMER (Physical Vascular Therapy), are used to optimise the performance and welfare of horses. The purpose of the present research was to investigate the impact of BEMER on body surface temperature using IRT in the distal parts of equine forelimbs. In addition, an ultrasonographic examination was performed to assess changes in the lateral palmar digital vein and artery diameters. Sixteen horses were divided into two equal groups: the active BEMER group (*n* = 8), which had BEMER boots applied to the distal parts of the forelimbs, and the sham group (*n* = 8), which had BEMER boots applied but without activation of the device. Body surface temperature and blood vessel diameter were measured before (BT) and just after (JAT) BEMER therapy. A third body surface temperature assessment was repeated 15 min after therapy (15AT). No significant temperature differences in body surface temperature were found in the study group at any timepoints after BEMER therapy for most of the measured areas. However, there was a significant increase in artery and vein diameter JAT indicated by ultrasonography examination. IRT did not identify changes in skin surface temperature after BEMER therapy at the distal parts of the forelimbs.

**Abstract:**

The study aimed to evaluate the impact of BEMER (Physical Vascular Therapy) on body surface temperature using infrared thermography (IRT) in the distal parts of the forelimbs in Thoroughbreds. The study tested the hypothesis that BEMER therapy leads to an increase in body surface temperature and blood vessel diameter in the distal parts of the forelimbs. The study involved 16 horses, split into 2 groups: active BEMER (*n* = 8) and sham (*n* = 8). The active BEMER group had BEMER boots applied to the distal parts of the forelimbs, whereas the sham group had BEMER boots applied without activation of the device. Both groups underwent IRT examination to detect changes in body surface temperature, followed by ultrasonographic examination to assess changes in vein and artery diameter before (BT) and just after (JAT) therapy. The IRT examination was repeated 15 min after BEMER therapy (15AT). There were no significant body surface temperature differences between BT and JAT in any regions of interest (ROIs) in either group. In the active BEMER group, the ROIs did not change significantly at 15AT, compared to the temperatures measured at BT (except for the hooves). At 15AT the temperature of all the ROIs (except the fetlock bone) dropped significantly in the sham group. In the ultrasonographic examination, there was a significant increase in vein and artery diameter in the study group JAT, whereas the sham group had a significant increase only in artery diameter JAT. These results suggest an effect of BEMER on stimulating blood circulation in the distal parts of the forelimbs in clinically healthy horses. IRT did not identify changes in skin surface temperature after BEMER therapy at the distal parts of the forelimbs.

## 1. Introduction

Infrared thermography (IRT) in equine practice is a fast, passive, non-contact and non-invasive physiological imaging method for monitoring body surface temperature [1,2]. IRT measures the infrared radiation emitted from a surface, with the radiant power detected allowing the calculation of the target surface temperature. The output, known as a thermogram, is expressed as an image that is colour-coded for temperature [3]. In conjunction with visual examination, thermograms can be analysed using thermographic software to derive pertinent temperature data from the image, including the average temperature within specified regions of interest.

The major influence on a horse’s body surface temperature is thermal conduction from the vascular network of superficial veins and arteries [4]. Because of their more superficial location, the temperatures of areas overlying the veins are normally higher than they are over the arteries, with temperatures varying and changing according to the thermoregulation of blood flow to the skin surface [5]. Dynamic changes in body surface temperature are additionally influenced by anatomical structures, the density and volume of subcutaneous and muscle tissue, as well as the characteristics of the hair coat [6].

Environmental factors also significantly influence cutaneous blood circulation and contribute to variations in body surface temperature [7,8,9]. Therefore, the effective utilisation of IRT requires a controlled environment and proper assessment of the horse’s physiological state, to control variability and the risk of errors in interpretation [6,10].

IRT is a useful supplementary tool in equine medicine for detecting and locating abnormal thermal patterns, characterised by local increases or decreases in body surface temperature [11]. It has been used to detect locomotion injuries in racehorses and to monitor their health status [6]. IRT has assisted in the diagnosis of a variety of orthopaedic injuries associated with the distal parts of the forelimbs, including tendinopathy, bucked shins [12,13], inflammation of the fetlock and carpal joint [14], laminitis [15] and podotrochleosis [16]. The thermographic patterns of healthy horses showing a high degree of symmetry between the left and right sides of the body [9,17], have also been used to improve diagnostic interpretation [7,18,19].

Areas of the cannon bone and fetlock joint from the medial aspect and the coronary band from the dorsal aspect present the highest body surface temperatures in the distal parts of the forelimbs [19]. The routes of the medial palmar vein/artery and medial digital vein/artery produce warm regions, as they are situated more superficially to the skin (between the cannon bone and the flexor tendons) in comparison to the lateral palmar vein/artery and lateral digital vein/artery routes. In contrast, areas away from major blood vessels appear cooler; these include the cannon bone, fetlock and pastern joint (both from the dorsal and palmar aspects) [20]. The lower temperatures in these areas are associated with the positions of the extensor and flexor tendons of the digit joints, which have poor blood supply. The coronary band is situated close to the major arterio-venous plexus [11,21] and is 1–2 °C warmer than the remainder of the hoof. The surface temperature of the hoof gradually decreases towards the ground [22]. According to previous studies, increased or decreased blood flow can be most reliably detected at the major blood vessels in the distal parts of the limbs, from the lateral and medial aspects [9].

IRT can play a role in evaluating therapeutic devices for equine physiotherapy. Previous studies have assessed the effect of laser therapy [23,24] and extracorporeal shock wave therapy [25] on body surface temperature in the distal parts of the forelimbs. All of these studies have demonstrated the utility of IRT in the assessment of temperature changes in response to therapy. Other studies have investigated the effects of a static magnetic field [26,27] and pulsed electromagnetic fields (PEMF) [28] on body surface temperature. The PEMF BEMER system (Physical Vascular Therapy) is a novel treatment that uses a pulsed electromagnetic field (10–100 μT) with a series of half-wave-shaped sinusoidal intensity variations (8–11 Hz and 28–31 Hz) to increase vasomotion and microcirculation for improved organ blood flow. In human studies, microscopy, computer image processing and laser reflection spectroscopy have indicated significant increases in the vasomotion of microvessels, the number of open capillaries and arteriolar and venular flow volume after BEMER therapy [29,30]. The specific biophysical and cellular mechanisms by which BEMER therapy promotes increased blood flow continue to be a central focus of ongoing research. In equine research, there have been reports following the use of this device demonstrating only a limited effect of BEMER [31,32], but no previous studies have measured surface temperature changes in response to treatment in distal parts of the limbs. This area is of particular interest in the assessment of performance horses as it is the most common site of injury or sub-clinical inflammation [33,34]. Monitoring vasomotor activity in the distal parts of forelimbs is crucial for detecting injuries or assessing rehabilitation processes. As such, this study evaluated the impact of BEMER therapy on body surface temperature using IRT on the distal parts of the forelimbs in clinically healthy racing Thoroughbreds. The study tested the hypothesis that BEMER therapy leads to an increase in body surface temperature and blood vessel diameter in the distal parts of the forelimbs.

## 2. Materials and Methods

The Animal Welfare Advisory Team at Wroclaw University of Environmental and Life Sciences approved the study design in compliance with Polish and European Union legislation on animal experimentation (no. 4/2024). The procedures used in this study were deemed not to cause pain, suffering, distress or lasting harm equivalent to or greater than that caused by the introduction of a needle (Article 1.5 of EU Directive 2010/63/EU).

### 2.1. Animals and Study Design

A blind, randomised, sham-controlled trial was undertaken at Wroclaw Horse Race Track in Partynice, Poland, in June 2023. The study included 16 clinically healthy Thoroughbreds (9 mares and 7 stallions) aged between 2 and 7 years without any clinical signs of injuries. All horses were at a similar level of regular training for flat racing, 6 days a week, with daily work under the saddle not exceeding 1 h, preceded by a warm-up in a walker for 20 min. The horses were accommodated in single boxes (3 m × 3.5 m) with straw within one stable. Prior to the study, the horses included in the study underwent visual and manual assessment by an equine clinician to detect any health issues. A routine physical examination of the musculoskeletal system was carried out by an experienced equine clinician to detect any clinical injuries. It involved an assessment of movement to ensure that the horses included in the study did not exhibit any lameness [35]. None of the horses were on any medications.

The horses were split into two equal groups with regard to age and gender: an active BEMER group (*n* = 8) including 4 mares and 4 stallions aged between 2 and 6 years old and a sham group (*n* = 8) including 5 mares and 3 stallions also aged between 2 and 6 years old. The active BEMER group had the BEMER boots device applied to the distal parts of the forelimbs, whereas the sham group had them applied without activation. Just before examination, all the horses were shaved for ultrasonographic examination, at the lateral aspect of the fetlock joint (blade size = 0.8 mm), where both of the vessels run superficially (at the level of sesamoid bones). An area of 4 cm^2^ was shaved consistently on all the horses. Horses from both groups immediately after the shaving underwent IRT examination before (BT) and just after (JAT) therapy to detect changes in body surface temperature in the distal parts of the forelimbs, followed by ultrasonographic examination to assess changes in the vein (lateral palmar digital vein) and artery (lateral palmar digital artery) diameter in the area of the fetlock joint. Additionally, the IRT examination was repeated 15 min after (15AT) BEMER therapy.

### 2.2. BEMER Physical Vascular Therapy

The horses were examined at rest, before daily exercise, in the corridor outside each horse’s box. Before the examination, each horse was taken out of the box for 20 min acclimatisation [5,7,19] in a closed stable without direct sunshine or an air draft. One hour before examination, the horses had their examination areas brushed [36,37]. The therapy was performed with BEMER boots (BEMER Int. AG, Triesen, Lichtenstein) applied to the distal part of both forelimbs, covering the area from the cannon bone to the hoof (Figure 1). The boots were put on the limbs with each of the three straps tightened equally, with the same procedure applied to each horse by the same person. The BEMER protocol consisted of a 15 min therapy programme that transmitted an electromagnetic signal of flux density 30–100 μTesla (the highest-level programme). During treatment and after taking off the boots for additional thermographic examination, each horse was held by a qualified person outside its box in the stable corridor. The sham group underwent the same procedure, but with the device turned off.

### 2.3. Infrared Thermography Examination

Body surface temperature was measured using a calibrated VarioCam HR infrared camera (uncooled microbolometer focal plane array, Focal Plane Array sensor size of 640 × 480, spectral range 7.5–14 μm, noise equivalent temperature difference of <20 mK at 30 °C, using the normal lens with IFOV of 0.57 mrad, measurement uncertainty of ±1% of the overall temperature range, InfraTec, Dresden, Germany) by the same non-blinded operator. The camera was set to have an emissivity of 1 for all readings [1]. Thermographic images of the distal parts of the forelimbs were recorded from both sides to include the lateral and medial aspects, which reveal far more detail of the distal limb vasculature than the dorsal aspects [6,19]. According to Soroko et al. [9], temperatures measured from the lateral and medial aspects are more likely to be more susceptible to influence by vasomotor changes than temperatures measured at the dorsal surface. Images of both forelimbs were taken from a distance of 1 m at a viewing angle of 90°.

The captured images were processed using software (IRBIS 3 Professional, InfraTec, Dresden, Germany) by one person to determine the average body surface temperature of the selected four regions of interest (ROIs): cannon bone, fetlock joint, fetlock bone and hoof (Figure 2). The shaved area with applied gel was avoided in temperature measurements [23,24]. The ambient temperature in the stable was 20 °C ± 3 °C with a humidity of 50%, measured with a TES 1314 thermometer (TES, Taipei, Taiwan).

### 2.4. Ultrasonographic Examination

Ultrasonographic evaluation of the lateral palmar digital vein and lateral palmar digital artery was performed using a 10 MHz linear transducer and a Draminski Blue ultrasound machine (Draminski^®^, Olsztyn, Poland). The examinations were performed by the same experienced ultrasonographer, who was blind to the treatment group allocated to each horse. A minimal amount of coupling gel at a stable temperature was applied to the shaved area, with minimal probe pressure applied to the tissue to avoid possible effects on vessel diameter [38,39]. After each examination, the gel was immediately removed from the skin with a paper towel. The vessels were visualised twice (BT and JAT) in transverse view to measure changes in their diameters.

### 2.5. Statistical Analysis

Compliance of the empirical distribution of continuous variables (body surface temperature and the diameter of arteries and veins) with the normal distribution was checked using the Shapiro-Wilk test. Student’s *t*-test for independent samples was used to compare the mean values of the variables between groups of horses. The means of variables in subsequent stages of the study were compared in pairs using Student’s *t*-test for paired samples, with the Holm-Bonferroni correction applied for multiple tests. A value of *p* < 0.05 was taken as statistically significant. To assess the correlation between changes in the diameter of the arteries and veins and changes in body surface temperature, the Pearson correlation coefficient (r) and Spearman’s rank correlation (rho) were estimated. Statistica v. 13.3 (TIBCO Software Inc., Palo Alto, CA, USA) was used for statistical analyses.

## 3. Results

There were no significant body surface temperature differences between the right and left or medial and lateral aspects of the distal parts of the forelimbs for both groups BT (*p* > 0.05). There were also no significant differences in body surface temperatures between the groups (*p* > 0.05).

In the active BEMER group, the ROI value differences between BT, JAT and 15AT (ΔT) in all ROIs JAT ranged from +0.1 °C to −0.3 °C; in the sham group, the changes ranged from +0.4 °C to −0.3 °C. None of the differences in ΔT for any of the regions were significant at timepoint 15AT. ΔT in all ROIs in the active BEMER group ranged from +0.4 °C to −0.2 °C, while in the sham group, they ranged from +0.5 °C to −0.1 °C. Differences in ΔT JAT and 15AT were not significant (*p* > 0.05).

The temperatures of all ROIs in the active BEMER group were significantly higher compared to the sham group at timepoints JAT (with the exception of the fetlock bone) and 15AT. The ROIs of the active BEMER group were on average 0.5 °C higher in the cannon bone, fetlock joint and hoof and 0.3 °C higher in the fetlock bone, compared to the sham group at 15AT. There were no significant body surface temperature differences in any ROIs between BT and JAT in either group. In the active BEMER group, the ROI temperatures did not change significantly (*p* > 0.05) at 15AT compared to the BT temperatures (except for the hooves). At 15AT, the temperatures of all the ROIs (except the fetlock bone) dropped significantly in the sham group (Table 1).

In the ultrasonography examination, there were significant differences in artery diameter between both groups at BT and JAT. The active BEMER group had significant differences in both artery and vein diameter at BT and JAT, while the sham group only had significant differences in artery diameter (Table 2).

In the active BEMER group, the strongest positive correlation was observed between changes in the diameters of the arteries (ΔAD) and veins (ΔVD) and changes in hoof temperature (ΔT_hoof_) (*p* < 0.05) JAT (Table 3).

## 4. Discussion

The study indicated that BEMER may not have resulted in a sufficient change in body surface temperature to be measurable by IRT. We found no significant temperature differences JAT in the active BEMER group. Research conducted on a static magnetic field has suggested that magnet-induced alterations in microcirculation depend on the initial vessel tone, indicating that the effect may vary depending on the condition of the treated tissue [40]. Previous studies based on the detection of the influence of magnetic blankets with IRT have indicated significant increases in the body surface temperature of horses’ backs after the treatment of the active and placebo groups [27,41]. The lack of temperature differences between both groups in the current study could be associated with a material insulation effect [42]. Blankets applied to muscular body parts, which have rich blood supplies and increased metabolic activity, are more likely to increase body surface temperature in response to insulation effects compared to distal limbs. It is possible that distal limbs require longer BEMER boot application in order to detect any effects on body surface temperature. It should also be considered that distal limbs are sensitive to ambient temperatures, which complicates the thermographic evaluation of body temperatures in this area. In a study undertaken by Soroko et al. [9], body surface temperature at the fetlock joint was strongly correlated with ambient temperature when the joint was viewed laterally and medially. Also, thermographic examination should be performed at an ambient temperature of approximately 20 °C with 20 min of acclimatisation for reliable results [5,19]. In the present study, the images were taken in a controlled environment under the recommended conditions to prevent artefacts. However, Tenley and Henson [7] found that only 19% of horses reached a thermographic temperature plateau after equilibration for 10 to 20 min, whereas 25% reached plateau values after 21 to 38 min of equilibration, and 56% required >39 min of equilibration. Moreover, the body temperatures of healthy horses are not constant over time, as physiological variations in body temperature occur [43]. Therefore, undetected temperature increments in body surface temperature in active BEMER group JAT could have been associated with diurnal variations in blood circulation, the reaction of horses’ bodies to magnetic fields, or adaptations to ambient temperature.

Our data align with prior findings [9,44] indicating that the temperature distribution between the two forelimbs of a healthy horse is symmetrical. Moreover, the temperatures in all the ROIs were the same for both groups BT. Interestingly, JAT and 15AT, there were significant temperature differences between the active BEMER group and the sham group. In addition, 15AT, the ROI temperatures for the sham group dropped significantly (except for the fetlock bone) compared to temperatures measured BT, while in the active BEMER group they stayed at the same level as BT (except for the hoof). This suggests that BEMER therapy could play a role in maintaining the same level of temperature in the distal parts of the forelimbs after therapy.

Previous studies have indicated that BEMER therapy stimulates regional blood flow [45,46]. In this study, the active BEMER group showed a significant increase in vein and artery diameter after BEMER therapy, but no significant increase in body surface temperature, which only partly confirmed the hypothesis of the study. The diameter of the lateral palmar artery and vein in the active BEMER group increased JAT by an average of 9% and 10%, respectively. There are no studies indicating what the percentage increase in the diameter of a blood vessel must be in order to observe an increase in body surface temperature with IRT. Perhaps the 10% increase in blood vessel diameter was insufficient to raise the local body surface temperature. In a previous study based on humans, it was shown that the relationship between changes in blood flow using venous occlusion plethysmography and skin temperature assessed with IRT is non-linear [47]. In another study based on seven human volunteers, the relationship between IRT and a laser Doppler imaging estimation of skin blood flow at identical locations was poor [48]. Another study elucidated the interrelation between changes in skin temperature and changes in skin blood flow, as measured by laser Doppler flowmetry and imaging. 17 healthy volunteers were studied upon immersion of both feet in water at 15 °C for 10 min, followed by body warming for 30 min. The researchers reported that temperature readings constituted an ambiguous estimate of skin blood flow [49].

According to Schmelz et al. [50] and Munce and Kenney [51], venodilatation can be a response to increased skin temperature and blood flow in the arteries, so the increased arterial diameter may reflect increased blood flow in the tissue together with venous dilatation. Similar results have been found in studies with humans [46], where, after BEMER therapy, blood flow, mixed venous oxygen saturation and relative venous haemoglobin, assessed by laser Doppler, showed a significant increase in the study group. Studies in which a static magnetic field was applied to the distal parts of horses’ forelimbs have shown significant increases in blood flow and metabolic activity in the metacarpus after therapy [52]. Conversely, a study by Steyn et al. [26] found that the application of magnetic wraps for 48 h did not increase blood flow to the portion of the metacarpus underneath the wrap.

In the sham group, we found a significant increment in artery diameter JAT, with no significant effects on body surface temperature JAT, and a significant drop in temperature at 15AT. A possible explanation for this finding is that veins are warmer than arteries, as they are metabolically active and are closer to the skin surface. The increase in artery diameter JAT could have contributed to the cooling effect, as well as the body surface temperature decrease for all ROIs 15AT. In the case of the veins, a significant increase in diameter was found in the active BEMER group JAT, which may have contributed to maintaining the same body surface temperature JAT and 15AT.

The current study has indicated a correlation between the lateral palmar digital vein and artery and hoof surface temperature changes in the study group. There are presently no studies that can confirm our results regarding the correlation between the body surface temperature of the hoof and the diameters of the arteries and veins. However, a possible explanation is that the coronary and laminar corium, just proximal to the hoof wall, are two of the warmest regions of the distal parts of the limb, due to the presence of the arteriovenous plexus [5,19]. Venous drainage from tissue with a high metabolic rate is warmer than venous drainage from normal tissue. In our previous studies based on laser therapy, we found a correlation between increments in the body surface temperature and the diameter of the cranial branch of the medial saphenous vein in the tarsus joint of healthy horses after laser therapy [23]. Opposite results were obtained in a study by Zielińska et al. [24], where there was no correlation between changes in body surface temperature and vein diameter in clinically healthy horses with pigmented and non-pigmented skin after laser therapy. A weak negative correlation was found between changes in vein diameter and body surface temperature in horses with clipped and non-clipped coats in the carpal joint [53].

Our study has a number of important limitations. A major one is the small sample size. The study included only 16 horses, which may have limited the statistical results. In addition, differences between individual horses in the two groups (recent training history/activity) could have had an effect on the results. The study should also have included a control group with the same protocol as both the active BEMER group and the sham group, but without applying the BEMER device. In addition, the ultrasonographic examination performed on the lateral aspect of the fetlock joint, which preceded the second thermographic examination AT, was performed on a shaved area, which could have contributed to an increase in the body surface temperature in both groups. The shaved area was avoided during temperature measurements, making this area inconsistent in terms of data analysis. In addition, ultrasonography should also have been performed 15AT to compare the results with body surface temperature. Finally, the study should also have included a long-term assessment of BEMER therapy on the treated area.

## 5. Conclusions

The study did not show any significant temperature differences in body surface temperature in the active BEMER group at any timepoints after therapy for most of the measured areas in the distal parts of the forelimbs. However, artery and vein diameter significantly increased in the active BEMER group after therapy. Therefore, the results suggest an effect of BEMER on stimulating blood circulation in the distal parts of forelimbs in clinically healthy horses. The study indicated that BEMER may not have resulted in a sufficient change in body surface temperature to be measurable by IRT. This survey highlights the need for further research into the benefits of BEMER therapy on equines in order to validate its efficiency. However, well-performed studies with larger sample sizes and longer applications of BEMER therapy are required for a more exact evaluation of the above-mentioned effects.

## Figures and Tables

**Figure 1 animals-14-01538-f001:**
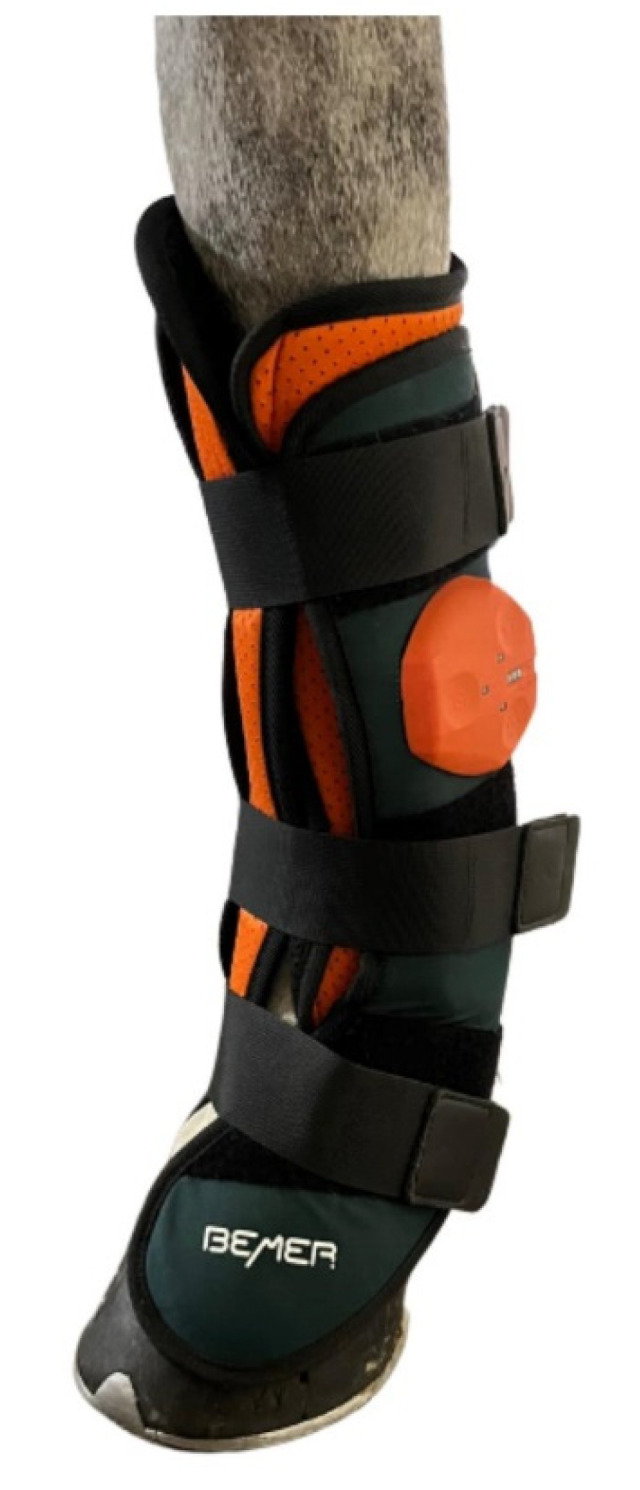
Example of BEMER physical vascular therapy boot applied to the distal part of the left forelimb.

**Figure 2 animals-14-01538-f002:**
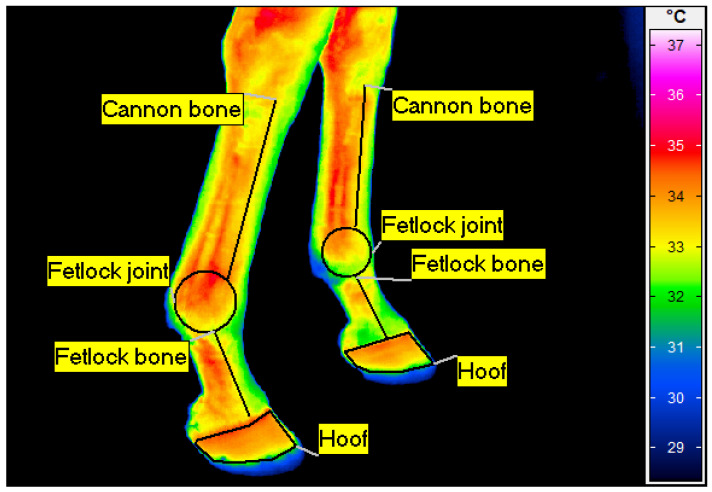
Example of a thermographic image of the distal parts of the forelimbs (lateral and medial aspects), with measured regions of interest indicated: cannon bone, fetlock joint, fetlock bone and hoof. Image taken before BEMER physical vascular therapy.

**Table 1 animals-14-01538-t001:** Basic temperature statistics in four regions of interest (ROIs) of the distal parts of forelimbs of both horses’ groups: active BEMER group (AB) and sham group (S) at three timepoints (BT—before therapy; JAT—just after therapy; 15AT—15 min after therapy) used in the study and the results of significance tests.

ROI	Group	BT	JAT	15AT	BT vs. JAT ^b^	BT vs. 15AT ^b^
Cannon bone	Active BEMER	33.8 ± 0.6	34.0 ± 0.6	33.9 ± 0.6	*p* = 0.134	*p* = 0.557
	Sham	33.7 ± 0.4	33.6 ± 0.6	33.4 ± 0.6	*p* = 0.265	*p* = 0.013 *
	AB vs. S ^a^	*p* = 0.449	*p* = 0.011 *	*p* = 0.002 **		
Fetlock joint	Active BEMER	33.7 ± 0.6	33.7 ± 0.6	33.7 ± 0.5	*p* = 0.718	*p* = 0.854
	Sham	33.5 ± 0.4	33.3 ± 0.7	33.2 ± 0.5	*p* = 0.108	*p* = 0.007
	AB vs. S ^a^	*p* = 0.210	*p* = 0.037 *	*p* < 0.001 ***		
Fetlock bone	Active BEMER	33.2 ± 0.6	33.2 ± 0.6	33.3 ± 0.5	*p* = 0.552	*p* = 0.313
	Sham	33.0 ± 0.5	33.1 ± 0.6	33.0 ± 0.5	*p* = 0.585	*p* = 0.965
	AB vs. S ^a^	*p* = 0.303	*p* = 0.311	*p* = 0.035 *		
Hoof	Active BEMER	34.1 ± 0.4	34.1 ± 0.5	33.9 ± 0.5	*p* = 0.691	*p* = 0.004 **
	Sham	33.9 ± 0.4	33.9 ± 0.5	33.4 ± 0.4	*p* = 0.465	*p* < 0.001 ***
	AB vs. S ^a^	*p* = 0.098	*p* = 0.041 *	*p* < 0.001 ***		

*—*p* < 0.05, **—*p* < 0.01, ***—*p* < 0.001, ^a^—Student’s *t*-test for independent samples, ^b^—Student’s *t*-test for paired samples with Bonferroni-Holm correction.

**Table 2 animals-14-01538-t002:** Means and standard deviations of the diameters of arteries and veins measured before therapy (BT) and just after therapy (JAT) in the active BEMER group (AB) and sham group (S) and the results of significance tests.

Group	Diameter BT [mm]		Diameter JAT [mm]		Artery BT vs. JAT ^b^	Vein BT vs. JAT ^b^
	Artery	Vein	Artery	Vein		
Active BEMER	2.0 ± 0.4	2.9 ± 0.6	2.2 ± 0.5	3.1 ± 0.7	*p* = 0.033 *	*p* = 0.038 *
Sham	2.7 ± 0.4	2.7 ± 0.3	2.8 ± 0.4	2.9 ± 0.3	*p* = 0.019 *	*p* = 0.061
AB vs. S ^a^	*p* = 0.004 **	*p* = 0.382	*p* = 0.011 *	*p* = 0.502		

*—*p* < 0.05, **—*p* < 0.01, ^a^—Student’s *t*-test for independent samples, ^b^—Student’s *t*-test for paired samples with Bonferroni-Holm correction.

**Table 3 animals-14-01538-t003:** The values of linear Pearson correlation coefficients (r) and Spearman’s rho rank correlation between changes in skin surface temperature in the ROIs and changes in the diameter of arteries and veins of the distal parts of the forelimbs.

	Active BEMER Group				Sham Group			
	ΔAD [mm]		ΔVD [mm]		ΔAD [mm]		ΔVD [mm]	
	r	rho	r	rho	r	rho	r	rho
ΔT_cannon bone_ [°C]	0.185	0.332	0.346	0.482	−0.478	−0.309	−0.553	−0.381
ΔT_fetlock joint_ [°C]	0.325	0.295	0.424	0.494	−0.278	0.155	−0.676	−0.086
ΔT_fetlock bone_ [°C]	0.141	−0.049	0.264	−0.062	−0.466	−0.154	−0.626	−0.295
ΔT_hoof_ [°C]	0.808 *	0.896 **	0.714 *	0.754 *	−0.131	0.077	−0.269	0.061

*—*p* < 0.05, **—*p* < 0.01.

## Data Availability

The data that support the findings of this study are available from the corresponding author [M.S.D.], upon reasonable request.

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
