# Peer review of "The Application of Infrared Thermography in the Assessment of BEMER Physical Vascular Therapy on Body Surface Temperature in Racing Thoroughbreds: A Preliminary Study"

_animals, 2024, doi:10.3390/ani14111538_

Round 1

Reviewer 1 Report

Comments and Suggestions for Authors

he current article seeks to identify the possibility that IRT can be used to understand the effects of applying therapy by BEMER on the local temperature and diameter of veins in the limbs, in ventral regions. 

Paper likte thar  may be important for the real understanding of the effects of BEMER as a physiotherapeutic treatment.

There are still few studies that have evaluated the effect of new equipment used as physiotherapeutic methods widely used by veterinarians during and after competitions and training for the recovery of horse athletes. Research carried out as this could help in understanding the effects of these equipment/types of treatments on different regions of the horses' body. 

As previously mentioned, tools like these help to understand the real effects of new methods or equipment in treatments for sick horses or for the recovery of horses post-training or during and after competitions. This article fills another gap. This type of research may or may not increase the best use of this equipment.

What further controls should be considered?  Possible improvements could be not only increasing the number of animals but also using animals in other competition conditions or equestrian disciplines, as this would mean an additional number of horses could come from other disciplines.

As a suggestion in the tables and due to the high number of abbreviations in them, we would like to suggest that these abbreviations be used as footnotes in the tables. It prevents the reader from getting lost among them.

The conclusions clearly reflect what was researched.

The references appropriate

Author Response

Responses to Reviewer 1 comments

Manuscript ID: animals-2948945, entitled “The application of infrared thermography in the assessment of BEMER physical vascular therapy on body surface temperature in racing Thoroughbreds: A Preliminary Study”

  1. Comment: The current article seeks to identify the possibility that IRT can be used to understand the effects of applying therapy by BEMER on the local temperature and diameter of veins in the limbs, in ventral regions. Paper likte thar may be important for the real understanding of the effects of BEMER as a physiotherapeutic treatment. There are still few studies that have evaluated the effect of new equipment used as physiotherapeutic methods widely used by veterinarians during and after competitions and training for the recovery of horse athletes. Research carried out as this could help in understanding the effects of these equipment/types of treatments on different regions of the horses' body.  As previously mentioned, tools like these help to understand the real effects of new methods or equipment in treatments for sick horses or for the recovery of horses post-training or during and after competitions. This article fills another gap. This type of research may or may not increase the best use of this equipment.

Response: Thank you for your comment.

  1. Comment: What further controls should be considered? Possible improvements could be not only increasing the number of animals but also using animals in other competition conditions or equestrian disciplines, as this would mean an additional number of horses could come from other disciplines.

Response: Thank you for your comment. That is a goal for our next study.

  1. Comment: As a suggestion in the tables and due to the high number of abbreviations in them, we would like to suggest that these abbreviations be used as footnotes in the tables. It prevents the reader from getting lost among them.

Response: Thank you for your suggestion, however all abbreviations ale included in the tables titles.

  1. Comment: The conclusions clearly reflect what was researched. The references appropriate.

Response: Thank you for your comment.

Reviewer 2 Report

Comments and Suggestions for Authors

Title, Simple Summary and Abstract: Title addresses only one measurement (IRT) technique to assess the effect of the BEMER therapy and it is not clear what the role of ultrasound plays in the project.

What supports your statement that IRT is useful to identify the effect or BEMER therapy by the time no significant body surface temperature difference is recorded during your study.

Line 44: This conclusion is not relevant to your study that is not directed towards defining what BEMER therapy does but to identify if IRT can be used to measure the effect of the BEMER therapy

Line 109 not sure you can really use references 31 and 32 to support your narrative in the context of BEMER therapy, using this is highly speculative and has no scientific validation based on associated research with the system

Reference 33: this study has identified no significant effect of the BEMER technique, it's citation in the way you present it is non appropriate

Reference 34: this study didn't include a control group and the effects of the BEMER judged significant are subjective clinical parameters, the objective measurements lacked to demonstrate any significant effect of the BEMER blanket.

You need to present these two references in a non-misleading way about their conclusions

Line 115 aligned with the comments about the title: you use two different modalities to assess the effect of BEMER therapy and this should be more obvious in the title of your paper

Line 121-126: not sure about the editorial requirement but international norms require a formal ethics approval

Fig 1: the left leg nomenclature needs editing as it mentions fetlock bone 2 times (upper one should be fetlock joint not fetlcok bone)

Line 191: what guarantees a standardised pressure to the area examined by ultrasonongraphy?

Line 302 should you rather refer the study group as treatment group to make the reading more clear?

Line 326 - 332 are these references really relevant to this discussion as they are relative to other physitotherapy techniques?

Line 335 was there a power calculation performed prior to define the size of your populations?

Are sentences starting line 351 and starting line 353 not contradicting each other?

it would be more appropriate to separate the two measurement techniques of your study and formulate a separate hypothesis for each so you can discuss them indepedently. This would however change the subject of the study and redirect it towards an assessment of the effects of the BEMER therapy rather then the way to measure its effect. There is a mix of subjects.

If you assume that your sample size is to small to provide robust dataset, you should propose this study as a pilot study.

Comments on the Quality of English Language

overall well written

Author Response

Responses to Reviewer 2 comments

 Manuscript ID: animals-2948945, entitled “The application of infrared thermography in the assessment of BEMER physical vascular therapy on body surface temperature in racing Thoroughbreds: A Preliminary Study”

Comment 1: Title, Simple Summary and Abstract: Title addresses only one measurement (IRT) technique to assess the effect of the BEMER therapy and it is not clear what the role of ultrasound plays in the project.

Response: Thank you for suggestion. The current manuscript has been prepared for the special issue: “Equine Musculoskeletal System: Advances and Clinical Applications of Diagnostic Imaging and Functional Research” therefore in the study we wanted to underline  the role of thermography in the detection of BEMER effects. Therefore the title has been modified together with aim of the study and hypothesis. The ultrasonography examination was only additional examination to indicate if vein and artery diameter correlate with increased body surface temperature.

Comment 2: What supports your statement that IRT is useful to identify the effect or BEMER therapy by the time no significant body surface temperature difference is recorded during your study.

Response: Thank you for your comment. This sentence from abstract and from the main text has been deleted.  

Comment 3: Line 44: This conclusion is not relevant to your study that is not directed towards defining what BEMER therapy does but to identify if IRT can be used to measure the effect of the BEMER therapy

Response: Thank you for that comment. The conclusion part has been modified and information about IRT deleted, lines:47-50.

Comment 4: Line 109 not sure you can really use references 31 and 32 to support your narrative in the context of BEMER therapy, using this is highly speculative and has no scientific validation based on associated research with the system.

Response: Thank you for that comment. The sentence with references number 31 and 32 have been deleted from the text.

Comment 5: Reference 33: this study has identified no significant effect of the BEMER technique, it's citation in the way you present it is non appropriate : Reference 34: this study didn't include a control group and the effects of the BEMER judged significant are subjective clinical parameters, the objective measurements lacked to demonstrate any significant effect of the BEMER blanket. You need to present these two references in a non-misleading way about their conclusions

Response:  The sentence at line has been modified 112-115.

Comment 6: Line 115 aligned with the comments about the title: you use two different modalities to assess the effect of BEMER therapy and this should be more obvious in the title of your paper

Response: Thank you for that comment. In this research we mainly focused on detection changes of body surface temperature in response to BEMER. Ultrasonography has been additional tool in investigate BEMER therapy and compare results with thermography. The sentence at line 118-120 has been modified.

Comment 7: Line 121-126: not sure about the editorial requirement but international norms require a formal ethics approval

Response: In accordance with the Experiments on Animals Act from January 15th 2015 (Journal of Laws of the Republic of Poland, 2015, item. 266), concerning the welfare of the animals used for research or teaching purposes, the restrictions defined in the law shall not apply to:

1) veterinary services as defined by the Act from December 18th 2003 concerning veterinary

practices (Journal of Laws from 2004, No. 11, item 95 as amended in item 3), as well as

agricultural activity, raising and breeding livestock according to the Animal Welfare Act, not

designed to carry out medical procedures;

2) clinical veterinary studies carried out according to Article 37ah-37ak of the Act from September 6th 2001 – Pharmaceutical Law (Journal of Laws from 2008, No. 45, item 271 as amended in item 4);

3) activity aimed at identifying animals;

4) capturing wild animals for biometric and systematic assessment;

5) veterinary procedures which do not cause pain, suffering, distress or permanent health

impairment equal to or more invasive than the insertion of a needle.

Hence, according to point 5 described above, veterinary procedures used in the methodology of our study did not require the approval of the Ethics Committee.

The ethical statement and exemption (no 4/2024) as a formal ethics approval was obtained from the Animal Welfare Advisory Team of Wroclaw University of Environmental and Life Science and presented to the editorial office of Animals.

The misleading sentence (“Ethical approval was granted without a formal application”) has been removed.

Comment 8: Fig 1: the left leg nomenclature needs editing as it mentions fetlock bone 2 times (upper one should be fetlock joint not fetlcok bone)

Response: Figure 2 (thermographic image) has been corrected.

Comment 9: Line 191: what guarantees a standardised pressure to the area examined by ultrasonongraphy?

Response: There is no possibility to standardised the pressure. To avoid variable pressure on the vessels each examination was performed by the same experienced ultrasonographer using a minimum pressure enabling visualization of the round vessel for diameter measurement.

Comment 10: Line 302 should you rather refer the study group as treatment group to make the reading more clear?

Response: Thank you for that comment, the sentence has been corrected instead study group there is active BEMER group Line: 318.

Comment 11: Line 326 - 332 are these references really relevant to this discussion as they are relative to other physitotherapy techniques?

Response:  There are no studies which can confirm our results regarding the correlation between body surface temperature of the hoof and the diameters of arteries and veins. Therefore at discussion point we wanted to present if there were found any correlations between body surface temperature and blood vessels using other physical devices.

Comment 12: Line 335 was there a power calculation performed prior to define the size of your populations?

Response: A temperature change of at least 0.5 degrees Celsius was considered a clinically important effect of the experiment. For the adopted significance level of alpha = 0.05 and test power 1 - beta = 0.8, the minimum number of horses in each group should be at least N = 16

Comment 13: Are sentences starting line 351 and starting line 353 not contradicting each other?

Response: Thank you for that comment. The sentences at lines 374-376, have been modified.

Comment 14: it would be more appropriate to separate the two measurement techniques of your study and formulate a separate hypothesis for each so you can discuss them indepedently. This would however change the subject of the study and redirect it towards an assessment of the effects of the BEMER therapy rather then the way to measure its effect. There is a mix of subjects.

Response: Look comment 1.

Comment 15: If you assume that your sample size is to small to provide robust dataset, you should propose this study as a pilot study.

Response: Thank you for that comment. The title of the research has been updated, preliminary study has been added to the title.

Reviewer 3 Report

Comments and Suggestions for Authors

This manuscript aimed to evaluate the impact of BEMER therapy on horse distal forelimb surface temperature using infra red thermography and blood vessel diameter, hypothesising that BEMER therapy would increase body surface temperature and blood vessel diameter, due to enhanced circulation of blood in the distal parts of the forelimbs.  This is an interesting question but there are a number of limitations with the study design and interpretation of the results.

The summary and abstract have similar limitations to the remains of the manuscript which are described below, so would be improved by addressing in this context.

Introduction: This is focussed mainly on IRT value.  There would be value to addressing options for assessing limb perfusion/circulation, including limitations and why this might be interesting to assess.  More information on the background to PEMF and practical application of the boots/rug being tested would help the reader understand the rationale behind the study.

Is both arterial and venous diameter representative of increase in circulation?  Increased arterial diameter may reflect increased blood flow, but venous dilation is more likely to represent poor venous return which is not acknowledged.

The objective stated is not what is concluded in the conclusion, where it is concluded that 'IRT was confirmed as being a feasible diagnostic modality for identifying body surface temperature changes in the distal parts of the forelimbs of horses' which was not tested in this study and should have been stated as an objective and tested in the methods if it was to be tested.

It is hypothesised that BEMER would increase body surface temperature and blood vessel diameter, due to enhanced circulation of blood in the distal parts of the forelimbs.  Enhanced circulation was not tested/confirmed so should be removed from the hypothesis.

Materials and Methods:

Horses - The inclusion/exclusion criteria should be clearly stated. Horses in race training are frequently carrying injuries or balancing repetitive loading with repair. Did included horses have any clinical history of injuries? Had this been excluded?  Were horses on any medications? Were they all training at the same level?

The horses in both groups were not paired by any means and there were no controls, so all differences could have been entirely due to differences between individual horses in the 2 groups (age, sex, size, shape compared to the boot design, activity, injury status/history, recent training history/activity etc).

Methods -

There is no repeatability reported to show the validity of any of the procedures or testing. This needs to be included to validate the methods used.

More description of the BEMER boots and how they were fitted to individual horses is required.

IRT requires very controlled conditions for repeatable results and there is considerable variation between horses. As the authors acknowledge in the discussion, there were considerable limitations with the methods used to control the environment, which gives me considerable concerns about the validity of the data. E.g.

What was the activity of the horses prior to testing? - it was before active exercise but what activity were the horses doing.  Is 20 mins long enough for acclimation to conditions prior to testing?  What was the influence of clipping as this is likely to have a reflect. Was the clipped area identical in size and location between horses?  How was this proved? 

What were the conditions in the corridor where treatment took place?  light/ activity/airflow?

where was the horse for the time after treatment and what it doing?  If not standardised, the validity of the 15min IRT assessment is questionable.

The US gel would have have a very significant impact on IRT so measurement of IRT after this is likely to be more influenced by this than BEMER PEMF or not. Unless this has been validated?

How was the pressure applied on US standardised?  If not standardised, this BV diameter is much more likely to be affected by pressure of the probe than blood flow. How repeatable were these results in testing prior to the study or within the study?  How repeatable was this between and within horses?

Unblinded IRT testing could have significantly influenced results as any differences detected were small.

Results: The findings are presented clearly

Discussion: The authors need to be careful to ensure that they address the limitations of the study as limitations and not as a justification for why differences were not detected between the 2 conditions, which is the impression from the first paragraph in particular. 

The authors are correct that these were significant limitations of the study in paragraph 1 and in the paragraph about limitations. However, acknowledging these fundamental limitations of the study is not sufficient justification to accept the results as viable, so it is important to discuss how these were addressed and whether the results can be justified.

Vein and artery diameter should be discussed in relation to the differences in factors influencing them.  Poor venous return (dilated vein) is different from good circulation and perfusion.  

Conclusions: these do not clearly relate to the objective or hypothesis (see comment in introduction) and overinterpret findings.  

The objective stated is not what is concluded in the conclusion, where it is concluded that 'IRT was confirmed as being a feasible diagnostic modality for identifying body surface temperature changes in the distal parts of the forelimbs of horses' which was not tested in this study and should have been stated as an objective and tested in the methods if it was to be tested.

This would be better rewritten in the context of the hypothesis and objective, and what was tested in this study. And relate to the more inconclusive nature of the findings than is currently projected in the conclusion.

Comments on the Quality of English Language

The English is adequate

Author Response

Responses to Reviewer 3 comments

Manuscript ID: animals-2948945, entitled “The application of infrared thermography in the assessment of BEMER physical vascular therapy on body surface temperature in racing Thoroughbreds: A Preliminary Study”

Comment 1: This manuscript aimed to evaluate the impact of BEMER therapy on horse distal forelimb surface temperature using infra red thermography and blood vessel diameter, hypothesising that BEMER therapy would increase body surface temperature and blood vessel diameter, due to enhanced circulation of blood in the distal parts of the forelimbs.  This is an interesting question but there are a number of limitations with the study design and interpretation of the results. The summary and abstract have similar limitations to the remains of the manuscript which are described below, so would be improved by addressing in this context.

Response: The main text abstract and summary have been updated according to below comments.

Comment 2: Introduction: This is focussed mainly on IRT value.  There would be value to addressing options for assessing limb perfusion/circulation, including limitations and why this might be interesting to assess.  More information on the background to PEMF and practical application of the boots/rug being tested would help the reader understand the rationale behind the study.

Response: New text has been added in the introduction part: 115-118.

Comment 3: Is both arterial and venous diameter representative of increase in circulation?  Increased arterial diameter may reflect increased blood flow, but venous dilation is more likely to represent poor venous return which is not acknowledged.

Response: Both arterial and venous diameter representative of increase of skin surface temperature measured with IRT. Ultrasound measurements were an additional tool for assessment of vessel response to BEMER therapy in correlation to increase of skin surface temperature. Venodilatation can be a response to the increased skin temperature, but also increase blood flow in the arteries, so the increased arterial diameter may reflect increased blood flow in the tissue together with venous dilatation, added information at lines: 320-323.

Comment 4: The objective stated is not what is concluded in the conclusion, where it is concluded that 'IRT was confirmed as being a feasible diagnostic modality for identifying body surface temperature changes in the distal parts of the forelimbs of horses' which was not tested in this study and should have been stated as an objective and tested in the methods if it was to be tested.

Response: Thank you for that comment. The conclusion part in the main text , abstract and summary has been corrected, line: 29-31; 47-50; 374-378. 

Comment 5: It is hypothesised that BEMER would increase body surface temperature and blood vessel diameter, due to enhanced circulation of blood in the distal parts of the forelimbs.  Enhanced circulation was not tested/confirmed so should be removed from the hypothesis.

Response: Thank you for that comment. Word circulation has been deleted from hypothesis : line 120-122. 

Comment 6: Horses - The inclusion/exclusion criteria should be clearly stated. Horses in race training are frequently carrying injuries or balancing repetitive loading with repair. Did included horses have any clinical history of injuries? Had this been excluded?  Were horses on any medications? Were they all training at the same level?

Response: Thank you for that comment. Horses didn’t have any clinical history of injuries, none of them was on any medications and all of them were on similar level of training. Text in methodology part have been updated, lines: 134-135, 143.

Comment 7: The horses in both groups were not paired by any means and there were no controls, so all differences could have been entirely due to differences between individual horses in the 2 groups (age, sex, size, /shape compared to the boot design, activity, injury status/history, recent training history/activity etc).

Response: Thank you for that comment, that information have been added in the discussion part where limitation of the study is described, lines: 247-350.  Information in the methodology part have been corrected about splitting horses to two groups lines: 144-147.

Comment 8: There is no repeatability reported to show the validity of any of the procedures or testing. This needs to be included to validate the methods used.

Response: Thermal imaging  software converts measured infrared radiation emitted from the analyzed region of interest (ROI) into temperature . The emissivity coefficient of the ROI and its distance from the camera are taken into account. Thermal image software calculate for each ROI its minimum temperature, maximum temperature, average temperature and standard deviation. The measurement correctness (no artifacts) is assessed by calculating the coefficient of variation (CV = SD/mean), which, with the correct selection of ROI (with uniform temperature distribution), should not exceed 3%. In each horse, the temperature of the same ROIs was measured four times: on the right and left limb and in the lateral and medial view. The horses studied were clinically healthy, therefore  assumption of symmetry seemed justified. The significance of differences between these four results was tested using the Student's t-test for paired variables. The average value of these four measurements was taken into account in further analysis.

Comment 9: More description of the BEMER boots and how they were fitted to individual horses is required.

Response: More explanation has been added 165-166.

Comment 10: IRT requires very controlled conditions for repeatable results and there is considerable variation between horses. As the authors acknowledge in the discussion, there were considerable limitations with the methods used to control the environment, which gives me considerable concerns about the validity of the data. E.g.

Response: Thank you for that comment. The examination was conducted in the controlled environment under the recommended conditions to prevent artefacts what was decried in details in methodology part.

Comment 11: What was the activity of the horses prior to testing? - it was before active exercise but what activity were the horses doing.  Is 20 mins long enough for acclimation to conditions prior to testing?  What was the influence of clipping as this is likely to have a reflect. Was the clipped area identical in size and location between horses?  How was this proved? 

Response: Thank you for that comment. Included horses were at rest before daily exercises,  line: 159. Horses had 20 minutes of acclimatization  outside of the box what is in the agreement with previous studies: Eddy et al. 2001; Turner 1991, Tunley and Hanson et al 2004- reference have been added line 160. All windows and doors in the stable were closed to maintain constant environmental conditions, line 161. The skin on the lateral aspect of the fetlock joint (at the level of sesamoid bones) was shaved with a small electric shaver with 2 cm wide blade. The shaved area was 2 cm x 2 cm and was the same for all horses, line 151. The thermographic examination was performed immediately after the area was shaved in each horse. New text has been added, line 150-152.

Comment 12: What were the conditions in the corridor where treatment took place?  light/ activity/airflow?

Response:  Information included at line 159-161.

Comment 13: where was the horse for the time after treatment and what it doing?  If not standardized, the validity of the 15min IRT assessment is questionable.

Response: During BEMER therapy and after taking off the boot for first thermographic examination horses stayed outside its box in the stable corridor. Information in methodology has been updated: 168-170.

Comment 14: The US gel would have a very significant impact on IRT so measurement of IRT after this is likely to be more influenced by this than BEMER PEMF or not. Unless this has been validated?

Response: Ultrasound was performed using a minimal amount of coupling gel at stable temperature. The first thermographic measurement was performed before the first ultrasound examination. Before BEMER therapy, the coupling gel was removed from the skin with a paper towel. After BEMER therapy the second thermographic measurement was performed before the second ultrasound examination. New text has been added, line 203-206.

Comment 15: How was the pressure applied on US standardised?  If not standardised, this BV diameter is much more likely to be affected by pressure of the probe than blood flow.

Response: There is no possibility to standardised the pressure. To avoid variable pressure on the vessels each examination was performed by the same experienced ultrasonographer using a minimum pressure enabling visualization of the round vessel for diameter measurement.

Comment 16: Unblinded IRT testing could have significantly influenced results as any differences detected were small.

Response: The initial statistical analysis for thermographic and ultrasonographic examinations  (calculation of descriptive statistics of the results for individual horses) was performed by a person (KD) who didn’t participate in the data collection and who, in the first phase, did not know which horse belong to active BEMER group or sham group.

Comment 17: Results: The findings are presented clearly

Response: Thank you.

Comment 18: Discussion: The authors need to be careful to ensure that they address the limitations of the study as limitations and not as a justification for why differences were not detected between the 2 conditions, which is the impression from the first paragraph in particular. 

Response: Thank you for that comment. Text has been updated line 290-291.

Comment 19: The authors are correct that these were significant limitations of the study in paragraph 1 and in the paragraph about limitations. However, acknowledging these fundamental limitations of the study is not sufficient justification to accept the results as viable, so it is important to discuss how these were addressed and whether the results can be justified.

Response: Thank you for that comment. This study was a first research on assessment of BEMER therapy in distal limbs with thermography. The thermographic examination was performed according to literature standards, therefore obtained body surface temperature results could be only accociated with horse physiology (what was explained in 1st paragraph) or with BEMER thearchy – possibly longer term treatment should be performed to indicate more reliable results. This study was a preliminary research which goal is to improve future research. 

Comment 20:  Vein and artery diameter should be discussed in relation to the differences in factors influencing them.  Poor venous return (dilated vein) is different from good circulation and perfusion.  

Response: Both arterial and venous diameter representative of increase of skin surface temperature measured with IRT. That ultrasound measurements were an additional tool for assessment of vessel response to BEMER therapy in correlation to increase of skin surface temperature. Venodilatation can be a response to the increased body surface temperature, but also increase blood flow in the arteries, so the increased arterial diameter may reflect increased blood flow in the tissue together with venous dilatation. New text has been added, lines 320-323.

Comment 21: Conclusions: these do not clearly relate to the objective or hypothesis (see comment in introduction) and overinterpret findings.  

Response: Thank you for that comment. Conclusion part has been updated, lines 373-379.

Comment 22: The objective stated is not what is concluded in the conclusion, where it is concluded that 'IRT was confirmed as being a feasible diagnostic modality for identifying body surface temperature changes in the distal parts of the forelimbs of horses' which was not tested in this study and should have been stated as an objective and tested in the methods if it was to be tested. This would be better rewritten in the context of the hypothesis and objective, and what was tested in this study. And relate to the more inconclusive nature of the findings than is currently projected in the conclusion.

Response: Thank you for that comment, The sentence: 'IRT was confirmed as being a feasible diagnostic modality for identifying body surface temperature changes in the distal parts of the forelimbs of horses' has been deleted form the conclusion part. New conclusions have been added basing on the hypothesis of the study.

Reviewer 4 Report

Comments and Suggestions for Authors

The paper covers interesting information on horse rehabilitation. However, the amount of the data should be underlined in the title as a preliminary report, also because the hypothesis tested was not confirmed. and there are many limitations of the study The statistical analysis is not the optimal one. It is not clear, why the analysis of variance with all factors that can influence the results,  was not used. The sex and age were only mentioned.  The presentation of the results could be better. the results coming from ultrasonographic equipment are not described and underlined enough. In all parts of the paper (abstracts, results) there is only one sentence about observed differences. it is not underlined and separated enough.

In all parts of the paper (methods, results), it should be more clearly stated what will be compared with what- which group with which group. However, better results should be presented after the proper analysis of variance with all factors. The same lack of clear results presentation is visible later in the results section.   The reader must concentrate strongly to understand what is presented. The paper is interesting with deep discussion, however, more structure is needed (perhaps paragraphs?). Please make the results and discussion section more clear and ordered. The discussion needs a stronger connection with the tested hypothesis. It is not clear why you suggest IRT as the equipment for identification of BEMBER therapy if there were almost no differences found between stages of investigation (before/after). Please underline your point of view. It is difficult to follow your point of view when you write that the first hypothesis was not confirmed and then suggest IRT as useful. You do not underline results between BT and 15AT.

 Conclusions should be re-written.  They must be connected with the results. Now you write that IRT is a feasible tool, but the hypothesis was not correct? How did you divide this tool as feasible? Write your point of view, as it is not clear from your work now.

Please write only this, what is coming from your research - the suggestion that BEMPER  maintains surface temperature - move to the discussion section.

In detail:

L 28, 43 – write the method of investigation

L 70 –control variability?

Please exclude the names of authors from the text (L 135,136, 177, 190)– there is a special place for contribution at the end of the paper.  

L 130- what about the weight of horses? Can it influence the results?

Figure  – please give the photo of the BEMPER boots on the horse to visualize the idea better way

Figure 1- change into fig.2

L 191 –what about the temperature of the gel?

L 209 –either ? please be more detailed to help the rider follow your text better

L 225 –space needed

L 235- please underline that these results are coming from ultrasonography, special paragraph?

L 299-300 – please write more detailed on your suggestion

Author Response

Responses to Reviewer 4 comments

Manuscript ID: animals-2948945, entitled “The application of infrared thermography in the assessment of BEMER physical vascular therapy on body surface temperature in racing Thoroughbreds: A Preliminary Study”

Comment 1:The paper covers interesting information on horse rehabilitation. However, the amount of the data should be underlined in the title as a preliminary report, also because the hypothesis tested was not confirmed. and there are many limitations of the study. The statistical analysis is not the optimal one. It is not clear, why the analysis of variance with all factors that can influence the results,  was not used. The sex and age were only mentioned.  The presentation of the results could be better. The results coming from ultrasonographic equipment are not described and underlined enough. In all parts of the paper (abstracts, results) there is only one sentence about observed differences. it is not underlined and separated enough.

Response: Thank you for that comment. Preliminary study has been added to the title. In the presented experiment, analysis of variance was not used because only two groups of horses were compared: those undergoing active BEMER treatment and sham group. Horses were selected for both groups in such a way that they were homogeneous in terms of gender and age. Both groups had BEMER boots applied. The only differentiating factor between the two groups was the activation of the device. Due to observed individual differences in vessel diameters and body surface temperatures in both compared groups and within each group, changes in these parameters after BEMER therapy were analyzed.

Parameters before treatment (BT)

Active BEMER

N = 8

Sham

N = 8

p

Gender:

1.000

Mares, n (%)

4 (50.0)

5 (62.6)

Stallions, n (%)

4 (50.0)

3 (37.5)

Age (years), Me [Q1; Q3]

3 [3; 4]

3 [2; 4]

0.793

                  Min - Max

2 - 7

2 - 6

Artery diameter (mm), mean ± SD

2.0 ± 0.4

2.7 ± 0.4

0.006

Vein diameter (mm), mean ± SD

2.9 ± 0.6

2.7 ± 0.3

0.294

Tcannon bone (°C), mean ± SD

33.8 ± 0.6

33.7 ± 0.4

0.449

Tfetlock joint (°C), mean ± SD

33.7 ± 0.6

33.5 ± 0.4

0.210

Tfetlock bone (°C), mean ± SD

33.2 ± 0.6

33.0 ± 0.5

0.303

Thoof (°C), mean ± SD

34.1 ± 0.4

33.9 ± 0.4

0.098

More information about ultrasonography results have been added in the text 330-323.

Comment 2: In all parts of the paper (methods, results), it should be more clearly stated what will be compared with what- which group with which group. However, better results should be presented after the proper analysis of variance with all factors. The same lack of clear results presentation is visible later in the results section. The reader must concentrate strongly to understand what is presented. The paper is interesting with deep discussion, however, more structure is needed (perhaps paragraphs?). Please make the results and discussion section more clear and ordered. The discussion needs a stronger connection with the tested hypothesis. It is not clear why you suggest IRT as the equipment for identification of BEMBER therapy if there were almost no differences found between stages of investigation (before/after). Please underline your point of view. It is difficult to follow your point of view when you write that the first hypothesis was not confirmed and then suggest IRT as useful. You do not underline results between BT and 15AT.

Response: BEMER leads to an increase in the diameter of arteries and veins, thus improving impaired microcirculation by enhancing the transport of oxygen and nutrients to tissues. Currently, the method for assessing the effectiveness of treatment involves measuring arteries and veins diameters with ultrasonography, which requires shaving of the examined body parts. The aim of the study was to evaluate the impact of BEMER therapy on body surface temperature using IRT the distal parts of the forelimbs, in clinically healthy racing Thoroughbreds. The study tested the hypothesis that BEMER therapy leads to an increase in body surface temperature and blood vessel diameter in the distal parts of the forelimbs. Changes (increase) in artery and vein diameters observed in the ultrasound examination should be accompanied by changes (increase) in the surface temperature of the distal forelimbs measured by thermographic camera immediately after therapy (JAT) and 15 minutes after therapy (15AT). In the study, the dependent variables were changes (increase) in artery and vein diameters and changes (increase) in surface temperature of the distal forelimbs in four regions of interest (ROI: carpal bone, fetlock joint, fetlock bone, and hoof) induced by BEMER therapy. In each horse, temperatures in each ROI were measured in two projections (lateral vs. medial aspect) and on both sides of the body. Since the differences between the results obtained in these four repetitions were found to be insignificant, as verified by paired t-test, the average temperature calculated from these four results was considered in further analysis. In both groups, the results of temperature increase measurements had a distribution close to normal and homogeneous variances, therefore, Student's t-test was used for comparisons. Temperature changes in each ROI were analyzed separately considering their location on the limb, blood supply, and hair length.

The discussion part  together with conclusions have been corrected. Lines: 312-316; 320-323; 357-359; 373-378.

Comment 3:  Conclusions should be re-written.  They must be connected with the results. Now you write that IRT is a feasible tool, but the hypothesis was not correct? How did you divide this tool as feasible? Write your point of view, as it is not clear from your work now.

Response: Thank you for that comment, conclusions have been re-written, lines 373-378

Comment 4: Please write only this, what is coming from your research - the suggestion that BEMPER  maintains surface temperature - move to the discussion section.

Response: The sentence that BEMER maintains surface temperature is left only in the discussion part.

Comment 5: L 28, 43 – write the method of investigation

Response:  Method has been added lines: 29-30; 45.

Comment 6: L 70 –control variability?

Response: minimize variability has been changed to control variability, line 73.

Comment 7: Please exclude the names of authors from the text (L 135,136, 177, 190)– there is a special place for contribution at the end of the paper. 

Response: All names have been deleted from methodology part.

Comment 8: L 130- what about the weight of horses? Can it influence the results?

Response: We didn’t weight horses, but considering that they were the same breed and were in the level of training, their weight had to be similar. We haven’t found any research which indicated that weight contributes to differences in body surface temperature and blood vessels  diameter in distal limbs.

Comment 9: Figure  – please give the photo of the BEMPER boots on the horse to visualize the idea better way

Response: The photo has been added.

Comment 10: Figure 1- change into fig.2

Response: The figure 1 has been changed to figure 2, line: 191,195.

Comment 11: L 191 –what about the temperature of the gel?

Response: Ultrasound was performed using a minimal amount of coupling gel at stable temperature. The first thermographic measurement was performed before the first ultrasound examination. Before BEMER therapy, the coupling gel was removed from the skin with a paper towel. After BEMER therapy the second thermographic measurement was performed before the second ultrasound examination. New text has been added, line 203 and 205-206.

Comment 12:  L 209 –either ? please be more detailed to help the rider follow your text better

Response: either has been changed to “for both groups”, line 222.

Comment 13: L 225 –space needed

Response: Space added , line 238.

Comment 14: L 235- please underline that these results are coming from ultrasonography, special paragraph?

Response:  Information has been added, line 248.

Round 2

Reviewer 2 Report

Comments and Suggestions for Authors

Comment 1 and response: Your response indicates that you have modified your original research to make it fit with the requirements and theme of the special issue. This is very concerning and explains why there is a confusion through the narrative between the evaluation of the usefulness of IRT to assess the effect of the BEMER therapy and the actual effect of the BEMER therapy

Line 34: as highlighted in your summary, your research hypothesis was to confirm that the BEMER therapy induces an increase in body surface temperature in parallel to the vasodilation you could confirm with the use of ultrasound, your conclusion is addressing only the latter and doesn't address enough that IRT is probably inadequate to assess the effects of the BEMER therapy as it doesn't induce a significant increase in BST. I'm not sure that the study highlights anything else as it hasn't investigated anything else. I you want to focus on the use of IRT, you need to stick to this focus and do not digress about what the BEMER therapy is actually doing.

I'm satisfied with grammatical edits and the references "trimmings"

Thank you for your clarification about ethics approval.

However, while the material and method, as well as results sections are acceptable, the subject of the discussion and conclusions is diverting from the evaluation of the benefit of IRT to assess the BEMER therapy, towards a dissertation about the putative effects of the BEMER therapy. These are two very different topics and these sections need more review.

Comments on the Quality of English Language

Line 21: write in addition not in addiction

Author Response

Comment 1: Line 34: as highlighted in your summary, your research hypothesis was to confirm that the BEMER therapy induces an increase in body surface temperature in parallel to the vasodilation you could confirm with the use of ultrasound, your conclusion is addressing only the latter and doesn't address enough that IRT is probably inadequate to assess the effects of the BEMER therapy as it doesn't induce a significant increase in BST. I'm not sure that the study highlights anything else as it hasn't investigated anything else. I you want to focus on the use of IRT, you need to stick to this focus and do not digress about what the BEMER therapy is actually doing.

Response: Thank you for that comment. We agree that we should focus more on the assessment of the efficiency of IRT rather than BEMER therapy. New paragraph about correlation between ultrasound examination and IRT with new references was added in the discussion. Lines: 321-335.

Comment 2:I'm satisfied with grammatical edits and the references "trimmings". Thank you for your clarification about ethics approval.

Response: Thank you for your comment.

Comment 3: However, while the material and method, as well as results sections are acceptable, the subject of the discussion and conclusions is diverting from the evaluation of the benefit of IRT to assess the BEMER therapy, towards a dissertation about the putative effects of the BEMER therapy. These are two very different topics and these sections need more review.

Response: The conclusions part and discussion  have been updated, lines: 30-31; 49-50; 280-281; 321-335.391-392.

Reviewer 3 Report

Comments and Suggestions for Authors

The authors have improved the manuscript by addressing a number of recommendations made. However, there remain a number of fundamental limitations with study design that have not been addressed and could have a serious influence on the results.  Without validation of the techniques and treatment groups it is difficult to accept this as a validated study design. Admitting that there are limitations to study design in the discussion is not sufficient to allow the study results to be claimed to be valid.   

In my opinion, the following concerns need to be further addressed for the study results to be valid.

1. the unpaired nature of the treatment and control groups which cannot be compared with confidence that differences are not related to individual horses rather than treatment.

2. Validation and repeatability of the ultrasound evaluation is not addressed. Only a reply to validation of thermographic technique has been addressed. It is acknowledged by the authors that the probe pressure was not standardised which cannot allow repeatable and validated US measurement.

3. The effect of clipping on skin temperature has not been validated and shown not to influence the results.  If this has been done, this needs to be added to the methods.

4. The effect of US gel application and removal on skin temperature has not been assessed and validated to show that this does not influence skin temperature.If this has been done, this needs to be added to the methods.

In the absence of description and confirmation of repeatability and validation for the techniques used, it is difficult to be confident to believe any results reported do represent what the authors are claiming.  If further validation and repeatability can be added and described, then it would be much easier to be confident in the reported findings.

Comments on the Quality of English Language

Some improvement in the English could be done.

Author Response

The authors have improved the manuscript by addressing a number of recommendations made. However, there remain a number of fundamental limitations with study design that have not been addressed and could have a serious influence on the results.  Without validation of the techniques and treatment groups it is difficult to accept this as a validated study design. Admitting that there are limitations to study design in the discussion is not sufficient to allow the study results to be claimed to be valid.   

In my opinion, the following concerns need to be further addressed for the study results to be valid.

Comment 1: . the unpaired nature of the treatment and control groups which cannot be compared with confidence that differences are not related to individual horses rather than treatment.

Response: Research horses to both groups were selected in such a way that they did not differ in gender, age or preparation for measurements. The only difference was the treatment (BEMER active group vs. sham group). Of course, individual differences between horses cannot be excluded. Therefore, the absolute values of temperature and arterial diameter measured before and after treatment were not analyzed, but their changes (increases).

Comment 2. Validation and repeatability of the ultrasound evaluation is not addressed. Only a reply to validation of thermographic technique has been addressed. It is acknowledged by the authors that the probe pressure was not standardised which cannot allow repeatable and validated US measurement.

Response:  Thank you for your comment. There was a risk that repeatability of the ultrasonographic examination was  limited , but couple of papers confirmed that that ultrasonographic examination is used in studies and clinical trials for blood vessel assessment, including the assessment of their diameter.:

Li, S., McDicken, W. N., & Hoskins, P. R. (1993). Blood vessel diameter measurement by ultrasound. Physiological Measurement, 14(3), 291–297.

Menzies‐Gow, N. J., & Marr, C. M. (2007). Repeatability of Doppler ultrasonographic measurement of equine digital blood flow. Veterinary Radiology & Ultrasound, 48(3), 281-285.

References have been added : 205

In addition each examination was performed by the same experienced ultrasonographer using a minimum pressure enabling visualization of the round vessel for diameter measurement.

Comment 3. The effect of clipping on skin temperature has not been validated and shown not to influence the results.  If this has been done, this needs to be added to the methods. 

Response: We agree that clipping has an impact on body surface temperature. Therefore we avoided the shaved area for thermographic assessment . Sentence “The shaved area was avoided in temperature measurements” is added at line 191.  

Comment 4. The effect of US gel application and removal on skin temperature has not been assessed and validated to show that this does not influence skin temperature. If this has been done, this needs to be added to the methods.

Response: Thermographic examination avoided area where the gel was applied, we followed the procedure as published in the previous papers: Godlewska et al. 2020; Zielińska et al. 2021

Reference have been added line: 191-192.

In the absence of description and confirmation of repeatability and validation for the techniques used, it is difficult to be confident to believe any results reported do represent what the authors are claiming.  If further validation and repeatability can be added and described, then it would be much easier to be confident in the reported findings

Reviewer 4 Report

Comments and Suggestions for Authors

The paper was corrected. The detailed information on horse groups supports simple statistics used in this preliminary study.

I suggest correcting the line -134-135, and changing the information: "without any clinical history of injuries" into "without any clinical signs of injuries". The horses are aged up to 7 years, so writing about history is too far.  

Author Response

Comment 1: I suggest correcting the line -134-135, and changing the information: "without any clinical history of injuries" into "without any clinical signs of injuries". The horses are aged up to 7 years, so writing about history is too far.

Response: Thank you for your comment, the information “without any clinical history of injuries" has been changed into "without any clinical signs of injuries", line 134.